# Evaluating Cross-Domain Text-to-SQL Models and Benchmarks

**Mohammadreza Pourreza**
University of Alberta
pourreza@ualberta.ca

**Davood Rafiei**
University of Alberta
drafiei@ualberta.ca

## Abstract

Text-to-SQL benchmarks play a crucial role in evaluating the progress made in the field and the ranking of different models. However, accurately matching a model-generated SQL query to a reference SQL query in a benchmark fails for various reasons, such as underspecified natural language queries, inherent assumptions in both model-generated and reference queries, and the non-deterministic nature of SQL output under certain conditions. In this paper, we conduct an extensive study of several prominent cross-domain text-to-SQL benchmarks and re-evaluate some of the top-performing models within these benchmarks, by both manually evaluating the SQL queries and rewriting them in equivalent expressions. Our evaluation reveals that attaining a perfect performance on these benchmarks is unfeasible due to the multiple interpretations that can be derived from the provided samples. Furthermore, we find that the true performance of the models is underestimated and their relative performance changes after a re-evaluation. Most notably, our evaluation reveals a surprising discovery: a recent GPT4-based model surpasses the gold standard reference queries in the Spider benchmark in our human evaluation. This finding highlights the importance of interpreting benchmark evaluations cautiously, while also acknowledging the critical role of additional independent evaluations in driving advancements in the field.

## 1 Introduction

Significant progress has been made in translating natural language text to SQL statements over the past few years. The execution accuracy on the hold-out test of Spider (Yu et al., 2018b)–a large-scale cross-domain text-to-SQL benchmark– has improved from 53.5 in May, 2020 (Zhong et al., 2020b) to 85.3 in March, 2023 (Pourreza and Rafiei, 2023). The exact set match accuracy, without considering database cell values, on the same benchmark and over the same period has im-

proved from 65.6 (Wang et al., 2019) to 74.0 (Li et al., 2023a). Measuring such progress is hinged on reliable benchmarks and evaluation metrics.

Two standard metrics for evaluating the performance in this domain have been *exact set match accuracy* and *execution accuracy*. The former measures if a model-generated SQL query lexically matches a reference SQL query, whereas the latter measures if a model-generated SQL query produces the same output as a reference query (§ 4).

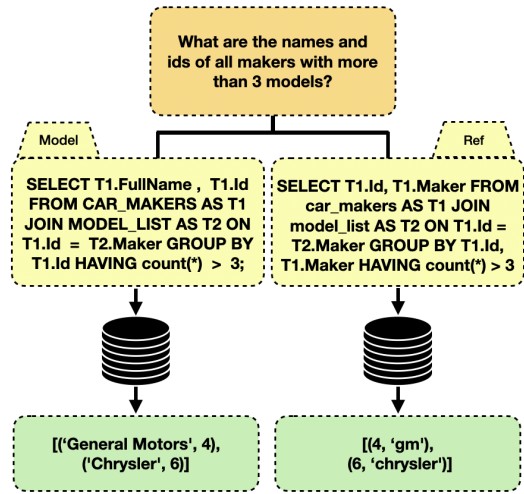

Figure 1: An example question with two correct SQL queries, each corresponding to a different interpretation. There is an ambiguity in schema mapping, with two different database columns describing the name.

Consider the example in Figure 1, which consists of a model-generated query (shown on the left) and a reference query (shown on the right). Both SQL queries return the id and name of makers that have more than 3 models. However, the model-generated query returns the column Full-Name, which gives the full name of a maker (e.g., "Ford Motor Company"), whereas the reference query given in the benchmark returns the column Maker, which gives the short common name of a maker (e.g., "Ford"). The model-generated query

fails an exact set match since the column names in the select clause are different. The query outputs are also different and the model-generated query fails the execution accuracy as well. The natural language utterance is not specific about the type of name to be returned, and a human evaluator tags both queries correct.

As the models improve, these types of failures make up most of the errors, and the performance metrics become less relevant, as shown in our evaluation. In particular, we re-evaluated all development set queries of Spider on which two top-performing models, one using a fine-tuned model (Scholak et al., 2021) and another using a large language model (Pourreza and Rafiei, 2023), failed. We found out that 25% of the queries generated by one model and 87% of the queries generated by the other model were indeed correct but were wrongly evaluated by the benchmark. For the same set of queries, our re-evaluation of the ground truth queries found 33% of the SQL queries incorrect, which was more than the number of incorrect queries generated by one of the models. This evaluation places one of the models above the ground truth queries in this re-evaluation.

We further re-evaluated two well-known benchmarks, Spider (Yu et al., 2018b) and Spider-DK (Gan et al., 2021b), and a newly released benchmark, BIRD (Li et al., 2023b), and found similar problems in all three benchmarks that affect the evaluation. Our evaluation reveals that 18% of the queries in the train sets and 20%-23% of the queries in the dev sets of these benchmarks are subject to ties in the dataset and which one of the tied rows are returned. This means a model-generated query will be deemed incorrect if it does not return the same row, among tied rows, as the ground truth query. This can severely impact the evaluation, especially when there is a tight race among models. Considering these observations, it is crucial to emphasize the significance of additional independent evaluations when utilizing these benchmarks. To enhance the evaluation process further, a potential solution is to incorporate multiple SQL queries as the ground truth, each representing a different interpretation that may be valid.

Our objective in this paper is to provide a comprehensive evaluation of existing Text-to-SQL benchmarks, underscoring the inherent issues they possess. We refrain from introducing a new dataset due to several considerations. First, addressing the identified issues by updating these benchmarks requires considerable human effort. Additionally, benchmarks in the Text-to-SQL domain, like Spider and BIRD, have holdout test sets used for official leaderboards and comparisons of text-to-SQL methodologies. We only have access to the development and training sets of these benchmarks, which limits our capability to alter the test sets. As a result, making changes only to the development and training sets would not completely address the benchmark's inherent problems, given that final performance is gauged using the problematic test sets.

## 2 Related Work

Limited research has been dedicated to assessing the reliability and effectiveness of Text-to-SQL benchmarks. The authors of SQL-PaLM (Sun et al., 2023) note in their qualitative analysis of their model that some queries, labelled as incorrect by execution accuracy, were considered correct by human annotators. Similarly, Lei et al. (2020) conduct an analysis highlighting the discrepancy between automatic evaluations and human annotations. They emphasize that certain queries produced by the models were labeled as incorrect SQL queries but human annotators labelled them as correct queries. Generally, a query that is equivalent (but not identical) to ground truth may be mistakenly classified as incorrect by automated evaluation metrics. Another study by Zhong et al. (2022) identifies limitations within the Spider benchmark, such as issues with ties and certain syntactic problems. Their analysis is primarily focused on a subset of Spider, without quantifying the extent or impact of these limitations or conducting an assessment of other benchmarks.

## 3 Text-to-SQL Benchmarks

Benchmarks have played a crucial role in advancing the field and providing a platform for evaluation. WikiSQL (Zhong et al., 2017) consists of over 24,000 tables from Wikipedia with SQL queries generated based on some predefined rules and templates. The queries in this dataset are considered easy since they are all single-table queries. Spider, introduced by Yu et al. (2018b), consists of 200 database schemas of which 160 schemas are published as train and dev sets and 40 schemas are kept hidden for testing. The queries are written on those schemas by Computer Science students

without using templates. This is considered a challenging dataset. Some other benchmarks are developed based on Spider, including Spider-Syn (Gan et al., 2021a), which replaces schema-related words with synonyms and eliminates explicit mentions between NLQ and schema, and Spider-DK (Gan et al., 2021b), which introduces rarely observed domain knowledge into the Spider development set. Other benchmarks include FIBEN (Sen et al., 2020), created for the financial domain and BIRD (Li et al., 2023b), which comprises 12,751 queries over 95 databases spanning 37 professional domains.

Our study in this paper focuses on cross-domain large-scale benchmark Spider, its variants Spider-DK and Spider-SYN, and a more recent cross-domain large-scale benchmark BIRD. The selection of these benchmarks stems from their resemblance to real-world datasets, which is a crucial factor in conducting comprehensive research and analysis. One notable advantage of these benchmarks is the availability of a large training set, which plays a pivotal role in training and fine-tuning large-scale models. The inclusion of a substantial amount of training data enables the development of more robust and powerful models that can better handle the complexities and nuances present in real-world databases.

## 4 Evaluation Metrics

The performance evaluation of text-to-SQL systems involves comparing them to a reference system, typically a gold standard set of known correct SQL queries. Generating a reference can be challenging due to multiple interpretations of natural language questions, while SQL queries are based on logic and tend to cover only one interpretation. Even if an interpretation is fixed, detecting if a model-generated query is equivalent to a reference query is challenging, due to the halting problem which is undecidable (Davis, 2004). Nonetheless, to assess progress, proxy measures of performance have been developed in the literature. As two such metrics, we review exact set match accuracy and execution accuracy in this paper.

Under *exact set match accuracy*, SQL queries are evaluated by matching the query clauses and components independently, such as the *select*, *where*, *having*, *group by*, and *order by* clauses. The matching is based on comparing columns and predicates, disregarding the ordering of columns and predicates. An exact

matching of literals can be challenging since predicates such as `nationality="Canada"` and `nationality="Canadian"` will not match. However, accurately generating those literals without accessing database content may not be possible. Under *exact set matching without values*, which is used in Spider (Yu et al., 2018b), a matching of literals is not required.

Two equivalent SQL queries can have different expressions and may not match under an exact set match. An alternative metric that can reduce the number of false negatives is the *execution accuracy*. Under execution accuracy, the equivalence between a model-generated query and a reference query is established if they both produce the same results on all possible databases instances (Yu et al., 2018a). While testing all instances is impractical, running queries on a subset of instances can help identify candidates that are not equivalent to the reference query. Although execution accuracy can detect queries that are equivalent but not identical, it may mistakenly identify queries as equivalent if they produce the same result on tested instances. Therefore, an effective execution-based evaluation requires finding instances that cover various edge cases and can detect queries that are not equivalent to the reference. Test suite accuracy (Zhong et al., 2020a), which is simply referred to as execution accuracy in Spider benchmark and in our work, aims to minimize false positives by evaluating queries on a carefully selected collection of database instances, known as a test suite. Nevertheless, an execution-based accuracy cannot capture all correct SQL queries, highlighting the limitations and the continued importance of human evaluation for reliable assessment.

## 5 Execution Accuracy Failures

A model-generated query can be correct but still fail the execution accuracy. We classify these failures into three categories: (1) failures due to ties in output, (2) ambiguity in schema matching, (3) wrong assumptions made about database content.

### 5.1 Failures Due to Ties in Output
SQL queries can lead to ties and a subset of the tied rows may be returned. The selection of tied rows can vary between queries and this can affect the execution accuracy. We identify a few sources for such ties, as discussed next, and study their impact on benchmark evaluations in Section 6. Table 1

provides a detailed breakdown of the number of queries that can potentially yield tied rows in both train and development set of Spider, Spider-DK, and BIRD benchmarks.

### 5.1.1 Top with Ties

Sometimes the query asks for top rows that satisfy some conditions (e.g., the student with the highest GPA, or the youngest student). When there is a tie for the top position, and the query in natural language is not specific on how the ties should be handled, the corresponding SQL query may return all ties or only one. This becomes a problem in evaluation if a model-generated query and the reference query treat the ties differently. Figure 2 provides a concrete example from the Spider dataset, illustrating this issue, where the reference SQL query in the benchmark fails to account for ties and returns only one of them using the LIMIT keyword.

### 5.1.2 LIMIT N

The problems associated with using the *LIMIT n* clause in SQL queries is not limited to the top position, as discussed above. The use of this clause is problematic for evaluation in general. Firstly, without an explicit ordering, the result of a SQL query is expected to be a set. Two equivalent (but not identical) queries can return the same set of results, each listed in different orders, but selecting *the first n* rows from one ordering will not necessarily match the same selection from a different ordering. Secondly, with query results sorted, there can be a tie on row *n* with multiple rows having the same values. The ordering among tied rows can vary between two queries, and so is the first *n* rows that are returned. All benchmarks studied in this paper (Spider, Spider-DK, Spider-SYN, BIRD) use the limit keyword and suffer from the aforementioned problems associated with ties.

### 5.1.3 GROUP BY

Many text-to-SQL benchmarks encounter a different type of issue associated with ties, particularly arising due to incorrect usage of non-aggregated columns in both the SELECT clause and the GROUP BY clause. Within the benchmarks, these ties manifest in two scenarios: 1) a column appears in the SELECT clause without being inside an aggregation function and without being included in the GROUP BY clause; 2) the SELECT clause contains a mix of aggregated and non-aggregated columns without utilizing a GROUP BY clause. In

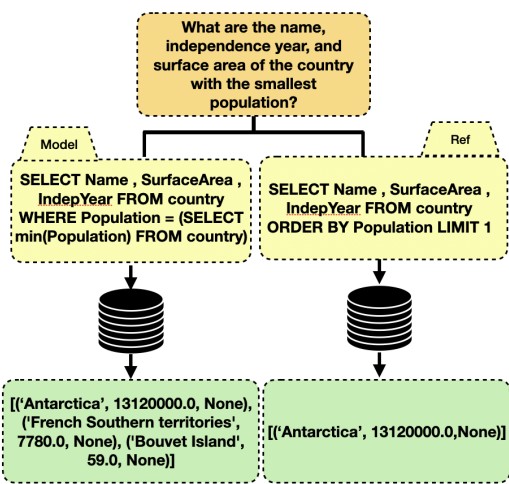

Figure 2: An example question that can have two correct SQL queries, each corresponding to a different interpretation. The SQL query on the left returns all tied values, while the SQL query on the right returns only one of the tied values.

both cases, multiple records can be associated with the same grouping column or aggregation value, whereas each group can only return one record. Some database systems including Oracle and DB2 prevent these cases by treating them as syntax errors. However, other database systems such as SQLite and MySQL take a more lazy approach (sometimes for efficiency reasons) and allow these cases to happen. Many text-to-SQL benchmarks follow SQLite syntax and suffer from this issue. The affected queries in our benchmarks were identified after migrating from SQLite to PostgreSQL, as detailed in Section 6.4, and checking for queries that failed during PostgreSQL execution. Figure 3, illustrates one example of such a problem from the Spider dataset.

### 5.1.4 ORDER BY

Another subtle ambiguity with tied values arises in queries where the SELECT clause incorporates the "distinct" keyword, paired with an ORDER BY clause referencing a column absent in the SELECT clause. Consider the exemplary query from Spider train set: SELECT DISTINCT district_name FROM district ORDER BY city_area DESC. The ordering of the output, as well as the result of a comparison with a reference query, becomes uncertain if a single 'district_name' value maps to multiple 'city_area' values. Similar to GROUP BY, the affected queries in the benchmarks were identified through a SQLite to PostgreSQL migration(§ 6.4).

| Benchmark | LIMIT 1 | LIMIT N | GROUP BY | ORDER BY | Total |
|---|---|---|---|---|---|
| **Dev set** | | | | | |
| **BIRD** | 255(16%) | 42(2%) | 20(1%) | 4(0.2%) | 321(20.86%) |
| **Spider** | 171(16%) | 10(0.9%) | 51(4.5%) | 2(0.2%) | 234(22.63%) |
| **Spider-DK** | 94(17%) | 2(0.3%) | 30(4.5%) | 2(0.3%) | 128(23.85%) |
| **Train set** | | | | | |
| **BIRD** | 1558(16%) | 211 (2%) | 23 (0.2%) | 4(0.04%) | 1792 (18.22%) |
| **Spider** | 989(14%) | 106(1%) | 254(3%) | 10(0.1%) | 1359(18.1%) |

Table 1: The number of SQL queries having a specific type of limitation together with the percentage on both development set and train set. The Spider-DK dataset does not have any training set.

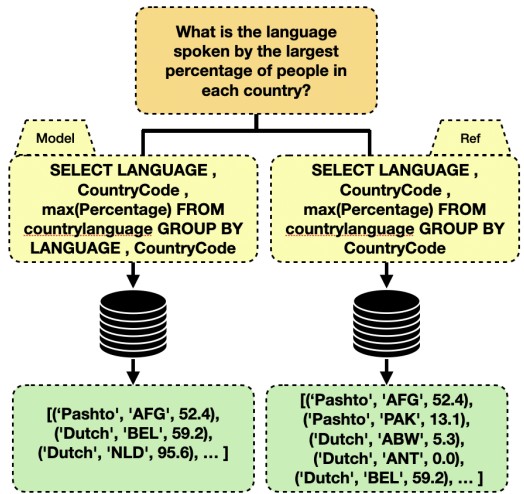

Figure 3: An example question that can have two correct SQL queries, each corresponding to a different interpretation. The SQL query on the left returns all languages of each country, each pair of country and language in a separate row, whereas the SQL query on the right returns one of tied values for the column LANGUAGE.

## 5.2 Ambiguity in Schema Matching

Schema matching refers to the task of establishing the correspondence between a natural language question and the tables, columns, and cell values in the database ((Cao et al., 2021; Pourreza and Rafiei, 2023; Wang et al., 2019; Li et al., 2023b). Ambiguities arise when there are multiple columns in the database that can represent the same semantic meaning, and the information needs of a query may be satisfied using any of those columns. As a result, there exist multiple SQL queries that can produce the correct answer, yet most benchmarks only provide one query among the many possible correct answers. Figure 1 illustrates an example question that can be satisfied by two different SQL queries, both of which are valid responses to the question at hand.

## 5.3 Wrong Assumptions on DB Content

Lastly, one type of limitation in text-to-SQL benchmarks stems from incorrect assumptions regarding cell values. It is common to make assumptions about database content and constraints when writing SQL queries, but those assumptions may not be supported by the database schema or content. This issue arises when the database content is created under assumptions that do not align with those in queries, leading to potential failures in the evaluation process. Text-to-SQL models often lack access to full database content due to limitations such as the context window problem and the inability to pass all cell values to the models for reasons such as privacy and cost. These models typically rely on the provided database schema and a selected sample of database rows to represent potential values (Pourreza and Rafiei, 2023; Liu et al., 2023; Rajkumar et al., 2022; Sun et al., 2023; Li et al., 2023a; Lin et al., 2020). Consequently, the assumptions made by these models may not align with the actual ground truth, resulting in SQL queries that are correct under the assumption made but do not match the reference query in the benchmark.

One observed case is when certain conditions (e.g., PetType='dog') are omitted from SQL queries due to the erroneous assumption that the condition holds for all rows in the database. Figure 4 exemplifies this issue using an example from the Spider dataset, where both queries yield the same answer on a specific database instance. However, changing the database values could result in failure, especially when evaluating performance using test-suite accuracy, which involves querying different database instances. Another case observed in the benchmarks is when the ground truth SQL queries assume a specific column has unique values, but in reality, that column does not possess that unique constraint. Figure 5 depicts an example of this

problem from the Spider dataset.

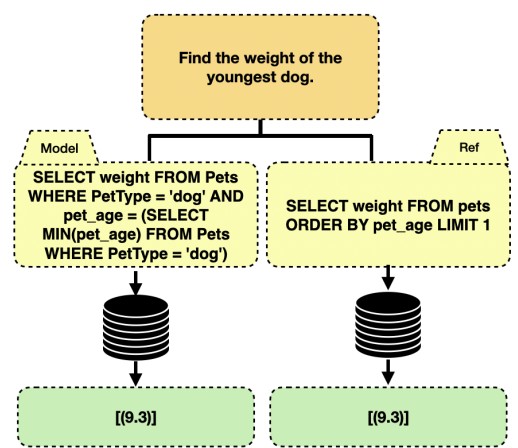

Figure 4: An example of a question and SQL pair with a wrong assumption on the cell values. The SQL query on the left does not make the same assumption.

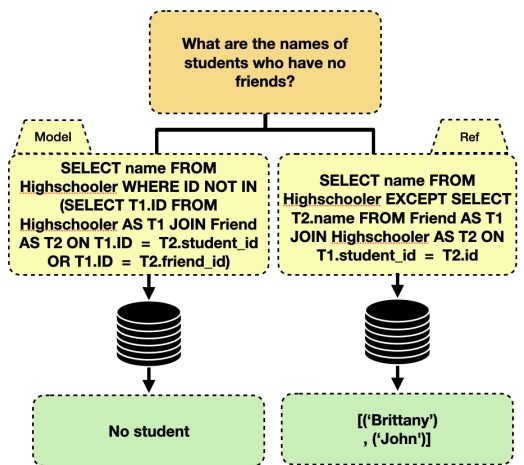

Figure 5: An example of a question and SQL pair with a uniqueness assumption on the "name" column, which is not supported by the schema. The SQL query on the left does not make the same assumption.

## 6 Experiments

To understand the extent at which the aforementioned problems affect the benchmarks, our evaluation and the ranking of the models, we conducted three types of evaluations on three benchmarks: Spider, Spider-DK, BIRD. Our findings here apply to the Spider-SYN dataset as well, which employs the same SQL queries as in the Spider dataset. For the same reason, we did not conduct a separate analysis of that benchmark.

### 6.1 Evaluation Through Query Rewriting

In this experiment, our focus is on ties and how a tie breaking strategy affects the benchmarks and our evaluation. This is done through query rewriting. Automating query rewriting faces inherent challenges, particularly when dealing with failures stemming from schema ambiguity, erroneous assumptions about the database content, and the ambiguity of natural language utterances. These challenges arise because there is no specific structure to address the failures systematically. Successful query rewriting in these cases necessitates a deeper understanding of table and column semantics to identify ambiguities and erroneous assumptions. In cases of ambiguity, human expertise is essential to disambiguate the context, as these situations often lack clear guidelines. Detecting erroneous assumptions often involves introducing new data to the database and meticulously reviewing and correcting failed queries on a case-by-case basis. Therefore, our efforts have been channeled towards rewriting queries concerning tied values, which adhere to a specific syntax structure, and the problems associated with the ambiguity in schema matching and wrong assumptions on database content are studied in the next section.

Many benchmark queries use "LIMIT 1" to find top rows that satisfy some conditions. If there are ties on top, one arbitrary row among ties is returned. An alternative is to return all ties. We rewrote all queries that used "LIMIT 1" to return all ties. This was done by introducing min() and max() aggregation functions within nested queries to accurately identify extreme values. An example of such rewriting is shown in Figure 2. Breaking ties for queries that used "LIMIT n" for $n > 1$ was not straightforward, and those queries were left unchanged.

For resolving ties introduced by an incorrect usage of GROUP BY in benchmark queries, we included all non-aggregated columns from the SELECT clause in the GROUP BY clause. For example, if the SELECT clauses included id and name, but the GROUP BY clause only included name, we added id to the GROUP BY clause. This change will not affect queries where there is a one-to-one mapping between id and name, but it will resolve the ambiguity when such mapping does not hold.

With these two changes, 16% to 20% of the reference queries in our benchmarks were affected. Under a perfect evaluation scheme, the accuracy should not be affected with these changes that sim-

| Benchmark | Affected Queries | Exec Acc | Set Match Acc |
|---|---|---|---|
| Spider | 206 (19%) | 92.3 | 81.6 |
| Spider-DK | 112 (20%) | 95 | 83.9 |
| BIRD | 252 (16%) | 96.87 | - |

Table 2: Performance of the revised SQL queries on the development set of the benchmarks.

ply resolve the uncertainty. Table 2 displays both the execution accuracy and the exact set match accuracy for the reference queries from the BIRD, Spider, and Spider-DK benchmarks after our modifications. It's important to highlight that the performance metrics provided in this table encompass the entire development set of these benchmarks, combining both modified and unaltered queries. For clarity, in the Spider dataset, out of 1034 queries, 206 were modified. The performance assessment took into account a mixed set of predicted queries: 206 that were adjusted and 828 that remained as originally presented. This culminated in an execution accuracy of 92.3 percent.

It can be noted that the execution accuracy is not as adversely affected as the exact set match accuracy. We hypothesize that this could be attributed to the absence of ties in the test data used for these benchmarks. An evidence of this is the following two queries, (Q1) `SELECT name, capacity FROM stadium WHERE average = (SELECT max(average) FROM stadium)`, and (Q2) `SELECT name, capacity FROM stadium ORDER BY average DESC LIMIT 1`, labelled as a correct match by the test scripts of Spider.

## 6.2 Human Evaluation

To gain a deeper understanding of the limitations within the benchmarks, we conducted an experiment focused on the widely-used text-to-SQL benchmark, the Spider dataset. Specifically, we evaluated two top-performing methods from the Spider leaderboard: DIN-SQL (Pourreza and Rafiei, 2023) and T5-large + PICARD (Scholak et al., 2021). This experiment involved running these methods on the development set of Spider, which comprised 1034 question-query pairs. From the results obtained, we extracted the questions for which both methods failed to produce a correct answer, based on the execution accuracy, resulting in 102 pairs. We then presented these questions, along with the SQL queries generated by the methods as well as the ground truth SQL queries (treating them the same as model-generated queries), to

two annotators [1] for labelling. The annotators had access to the database schemas and were tasked with identifying the queries they deemed correct for each question, without knowing which model generated which query or if the query was from the ground truth queries. Annotators could also create databases and validate queries, ensuring a thorough evaluation.

Following our initial labelling process, we wanted to minimize the potential impact of human errors in our evaluation. For this, we identified queries with inconsistent labels among the annotators and presented them to the annotators. Each annotator was asked to provide an explanation for their assigned labels. In the final stage of evaluation, each annotator was presented the inconsistent queries and the explanations provided by the other annotator. They were then asked if they would revise their labels based on this additional information. The results of this experiment are presented in Table 3. This table presents the outcome of human evaluation on a sample of 102 queries that both DIN-SQL and T5+PICARD methods were deemed incorrect in terms of execution accuracy. SQL experts conducted this evaluation, with 81.6% of these queries judged as correct for DIN-SQL, and only 25.5% for T5+PICARD. Notably, among the reference queries, only 67.3% were deemed correct. Even after the second round of annotation, a few queries (more specifically, four question-query pairs) still exhibit inconsistent labeling by the annotators. The main challenge with these particular pairs is the inherent ambiguity in the questions or the subjectivity of interpretations, which leads to a lack of a definitive answer. Figure 6 demonstrates one example of such a question with two possible SQL query as answers.

An intriguing observation emerged from this experiment: the DIN-SQL method, powered by GPT-4, produced the highest number of correct answers, surpassing even the ground truth SQL queries. This finding sheds light on the limitations of the current benchmarks and raises doubts about the reliability of current leaderboards and performance metrics.

## 6.3 Error Analysis of Human Evaluation

We performed an error analysis of the SQL queries that were labelled as incorrect in our human evaluation to better understand the error types and causes and to provide insights into areas for improving the

---

[1]The human annotators are the authors of this paper.

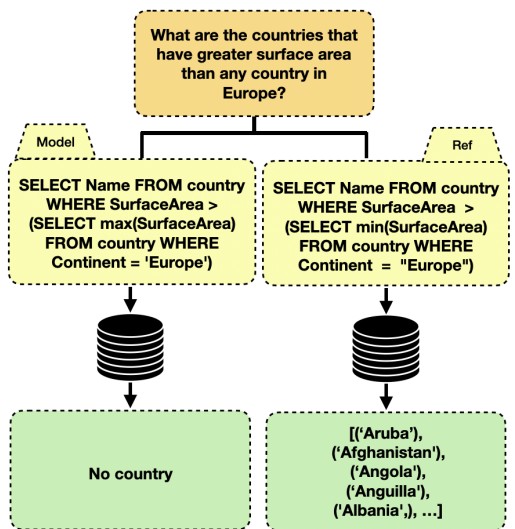

Figure 6: An example of a question with two possible SQL queries as the answers. Both of these SQL queries are correct under different interpretations.

| Method | Acc | Incon |
|---|---|---|
| **DIN-SQL** (Pourreza and Rafiei, 2023) | **81.6** | 4 |
| **T5-large + Picard** (Scholak et al., 2021) | 25.5 | 4 |
| **Ground Truth** | 67.3 | 4 |

Table 3: Accuracy of the SQL queries generated by two methods and the ground truth SQL queries based on human evaluation. In four cases, the two annotators did not agree on a label even after a second round.

ground truth SQL queries. Additionally, we compared the errors in ground truth queries with those of fine-tuning and prompting approaches. The identified errors, categorized into five groups, are briefly discussed next. The distribution of SQL queries across these groups is depicted in Figure 7.

**Schema** The primary issue responsible for the majority of errors, affecting both the reference SQL queries and the two methods, is the incorrect usage of schemas, which arises when the SQL query utilizes incorrect tables or columns to answer the given question. These errors indicate ambiguities in the database schema and/or questions, as discussed in Section 5. Notably, the reference set shows the least number of errors, which is closely followed by DIN-SQL.

**Condition** The second-largest group of errors observed pertains to the usage of incorrect conditions within the SQL queries. Unlike the schema

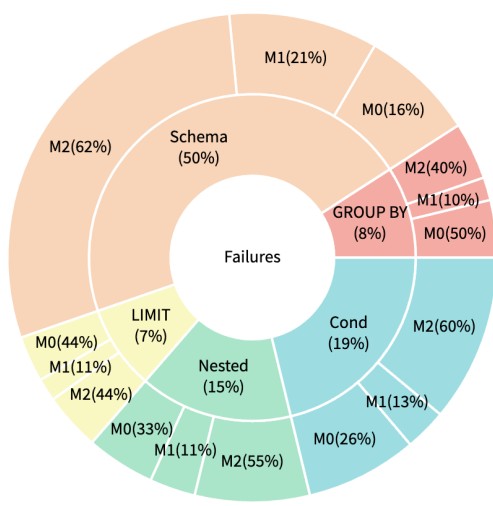

Figure 7: Distribution of SQL queries across error groups for the two models being evaluated and the ground truth. M0 refers to SQL queries in the reference (ground truth) set, M1 refers to the DIN-SQL method, and M2 refers to T5+PICARD.

group, where the tables and columns were incorrect, in this group, the correct tables and columns are used, but the conditions in the WHERE clause are erroneous. This error primarily manifested in the queries generated by the T5-PICARD method, but was also present in the reference set. The T5 model's tendency to introduce additional columns or omit necessary conditions could be attributed to its smaller size relative to larger models like GPT-4, limiting its grasp of intricate SQL syntax.

**Nested** The source of this problem is using a non-unique column for the nested SQL query, as also discussed in Section 5. Figure 5 shows an example of such an error in a SQL query. This error was more common in the SQL queries provided in the reference set as well as those of T5-PICARD.

**GROUP BY** This category includes queries that incorrectly used GROUP BY, resulting in ambiguity or uncertainty in the result as discussed in Section 5. Notably, the reference set showed the largest number of errors, closely followed by the fine-tuned T5-PICARD. DIN-SQL exhibited the least number of errors.

**LIMIT** As highlighted in Section 5, one of the error scenarios involves not properly handling ties when using the LIMIT keyword. The DIN-SQL method demonstrates a lower incidence of this type of error, attributed to its prompting nature. Conversely, T5-PICARD exhibits identical performance to the ground truth in this particular case.

| Benchmark | SyntaxErr | UndFunc | UndCol | Order By | Group By |
|-----------|-----------|---------|--------|----------|----------|
| Spider | 4 | 69 | 211 | 2 | 51 |
| Spider-DK | 134 | 62 | 80 | 2 | 30 |
| BIRD | 5 | 103 | 1 | 4 | 20 |

Table 4: Breakdown of SQL errors observed in Spider, BIRD, and Spider-DK, following migration to PostgreSQL.

## 6.4 Standard SQL validation

We undertook an extensive review of the development set of Spider, BIRD, and Spider-DK benchmarks through the lens of standard SQL validation. The objective was to identify some of the problematic queries discussed in Section 5 and assess the portability of the benchmarks. As part of this analysis, we migrated the databases and queries of these three benchmarks from Sqlite to PostgreSQL. Our decision to use PostgreSQL, a widely recognized RDBMS, stemmed from its rigorous adherence to SQL standards. Following the migration, we executed every query from the development set on these PostgreSQL databases, with a keen focus on identifying queries that failed during PostgreSQL execution. Table 4 provides a breakdown of queries by error type across all three benchmarks. Notably, errors such as UndefinedColumn, SyntaxError, and UndefinedFunction emerge due to the different SQL formats supported by Sqlite and PostgreSQL. These variances necessitate adjustments to make the queries compatible with PostgreSQL standards. For instance, the Spider dataset frequently showcases errors stemming from PostgreSQL's strict typing conventions. While SQLite allows for comparisons of int with text, PostgreSQL does not. Also, some queries run into problems because of SQLite-exclusive functions, such as strftime and iff, or because PostgreSQL interprets literals in double quotations as column names.

The two other types of failures, group by and Order by, included queries that introduced ambiguities to the benchmarks, as discussed in Section 5. It should be noted that these benchmarks present a range of issues that are not solely confined to syntax. Challenges related to wrong assumptions on DB content and ambiguities in schema matching are notably pervasive.

## 7 Discussion

Our analysis (§ 6.1) reveals the limitations of major text-to-SQL benchmarks, highlighting the fact that even with a perfect model, achieving a perfect accuracy on these benchmarks is not possible. The accuracies presented in Table 2 serve as a lose upper bound for the achievable accuracy by models. It is lose because our rewritings were unable to address cases that required manual intervention to reconstruct a correct query. Thus, the upper bound is expected to be lower considering other issues such as wrong assumptions on the database content and ambiguity in schema matching.

Our human evaluation (§ 6.2) further supports our claim and provides more insight into the limitations within one of the benchmarks studied. The results in Table 3 demonstrate that prompting methods, such as DIN-SQL, are less affected by the inherent limitations of the training set in the benchmarks. However, they are not fully immune because of the few-shot input-output demonstrations that are taken from the train set. On the other hand, fine-tuned approaches, such as T5+PICARD, perfectly mirror the distribution of errors seen in the ground truth queries for types nested, LIMIT, and GROUP BY. The largest number of wrong queries in schema and condition classes belong to our fine-tuned model, due to inability of the model to generate correct SQL queries.

## 8 Conclusions

The reliance on standard text-to-SQL evaluation metrics, namely exact set match accuracy and execution accuracy, has become less reliable as the model performance approaches human-level performance. Our work is the first to systematically study the limitations of these metrics and benchmarks through both human evaluation and query rewriting. Our re-evaluation of well-known benchmarks (Spider, Spider-DK, and BIRD) uncovers common systematic issues that affect the evaluation process and performance estimates, revealing that a significant portion of queries in the train and dev sets are impacted by these issues. Incorporating multiple SQL queries as the ground truth and representing different interpretations of queries offer a promising solution to enhance the evaluation process and achieve a more comprehensive and accurate assessment of Text-to-SQL models.

## Limitations

In this study, our focus was primarily on cross-domain text-to-SQL benchmarks and models. The failure cases identified in this domain are likely to be present in other domain-specific text-to-SQL benchmarks and models as well. It is essential to conduct further analysis to identify specific failure cases within domain-specific benchmarks and models.

Furthermore, it is worth mentioning that our work has a limitation regarding the analysis of failure cases that lack a specific structure and require manual effort for detection. Identifying and addressing such problems necessitates extensive work. The purpose of our study was to highlight these failure cases; a more in-depth analysis of their prevalence can provide a clearer understanding of their impact on the overall performance of text-to-SQL systems.

## Ethics Statement

In this paper, we acknowledge the importance of ethical considerations in conducting and presenting our research. We affirm our commitment to comply with the ACL Ethics Policy and adhere to ethical guidelines and principles throughout the entire research process.

We have taken necessary measures to ensure the privacy, confidentiality, and consent of individuals or entities involved in our data collection, experimentation, and analysis. Any personal or sensitive information used in this study has been appropriately anonymized and safeguarded.

Furthermore, we have made efforts to minimize any potential biases and discrimination in our research design, data selection, and interpretation of results. We have strived for transparency, accuracy, and fairness in reporting our findings, and we have provided appropriate citations and acknowledgments to give credit to the work of others.

By including this ethics statement, we aim to demonstrate our dedication to conducting research with integrity, respecting ethical principles, and contributing to the responsible advancement of knowledge in our field.

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
