# OpenReview forum: "Evaluating Cross-Domain Text-to-SQL Models and Benchmarks"
_EMNLP/2023/Conference — EMNLP 2023 Main_

### Official Review · Reviewer_5LcL · 2023-08-03

**Soundness:** 4

**Excitement:**

3: Ambivalent: It has merits (e.g., it reports state-of-the-art results, the idea is nice), but there are key weaknesses (e.g., it describes incremental work), and it can significantly benefit from another round of revision. However, I won't object to accepting it if my co-reviewers champion it.

**Paper Topic And Main Contributions:**

The authors study the limitations of text-to-SQL evaluation metrics and benchmarks through both human evaluation and query rewriting. They also summarize issues that affect the evaluation process and performance estimates.

**Reasons To Accept:**

The workload of this paper is quite heavy and the work is meaningful.

**Reasons To Reject:**

-They point out the issues with the existing evaluation methods, but not propose a new method to assess text-to-SQL models.

**Reproducibility:**

N/A: Doesn't apply, since the paper does not include empirical results.

**Reviewer Confidence:**

3: Pretty sure, but there's a chance I missed something. Although I have a good feel for this area in general, I did not carefully check the paper's details, e.g., the math, experimental design, or novelty.

---

> ### Author Rebuttal · Authors · 2023-08-26
>
> Thank you for taking the time to review our paper and for your valuable feedback. We appreciate your insights regarding the evaluation methods.
>
> In addition to drawing attention to the issues within existing benchmarks, we proposed a novel approach to address the evaluation challenges in text-to-SQL models. Specifically, we suggested using multiple SQL queries as references, which can better capture the inherent ambiguities present in natural language questions. This approach reflects the complexities of real-world scenarios, where databases often contain numerous tables and columns with various interpretations.
>
> We believe that by incorporating multiple SQL queries as references, researchers and authors can ensure a fairer and more comprehensive assessment of text-to-SQL methods. This approach enables a more accurate evaluation, considering the diverse interpretations that users might have when querying databases.

---

### Official Review · Reviewer_FJpx · 2023-08-04

**Soundness:** 4

**Excitement:**

4: Strong: This paper deepens the understanding of some phenomenon or lowers the barriers to an existing research direction.

**Paper Topic And Main Contributions:**

The paper provides an in-depth analysis of SQL queries in Text-to-SQL benchmarks (Spider and variants) and demonstrates that systemic semantic aspects of the SQL queries make the benchmark ambiguous or fragile in about 10% to 20% of the samples in the dataset.

The specific analysis relates to:

1. Aggregates (group by) where the list of fields that are aggregated does not include all selected fields.  (For example, "select id, name from T1 group by name").

2. Extrema queries (min, max) implemented through LIMIT (TOP n) - in these cases, depending on available data in the tables, TIEs may change the returned values.

3. Ambiguity in schema matching - when multiple columns (sometimes in different tables) may correspond to the focus of a question - this leads to systemic ambiguity in interpreting the meaning of the question.

4. Wrong assumptions on db content - for example, assuming that all rows in a table satisfy a condition (which may even be true given a snapshot of the content) but which does not correspond to a generalizable query.  (Example with "petType = 'dog').

Experiments are run to assess the impact of these phenomena in execution match metrics for Spider variants (Spider, Spider-Syn, Spider-DK, BIRD).  They show that up to 20% of the samples may be affected by these considerations - and that models that "overfit" to the interpretation embodied in the Spider gold SQL fail to produce correct execution results when the query is disambiguated or simply modified syntactically in a way that should be neutral to semantics.

These factors are shown to lead to "leaderboard reversals" - where some models rankings are changed when considering query rewriting.

**Questions For The Authors:**

I mostly derive the questions from the points raised in the "Reject" section above:

A. Please clarify the interpretation of the experiment reported in Table 2 - also provide examples (in Appendix if necessary) of the min/max/top transformation.

B. Please clarify the interpretation of the experiment with human evaluation reported in Table 3 - confirm these are accuracy figures over 102 samples selected to demonstrate the fragility of original Spider queries.

C. What is your recommendation on how to improve Text-to-SQL datasets based on the analysis reported in the paper?  You mention adding multiple versions of SQL for the same question - but these SQL queries could return different result sets - is this your recommendation? Or would you prefer to disambiguate problematic cases based on inter-annotator agreement?

[Post rebuttal]
I acknowledge good answers to these questions.

**Reasons To Accept:**

A. Good empirical analysis of popular text-to-SQL datasets showing their limit and calling for prudence when interpreting model accuracy.

B. Empirical results demonstrating the potential level of noise in the dataset caused by systemic problems supported by human analysis.

C. Suggestion to include multiple queries per question to address question ambiguity is well taken and will improve best practice.

**Reasons To Reject:**

A. The analysis of the SQL "quirks" indicated in the paper relies on non-standard SQL implementation of SQLite which is used in Spider (mostly for convenience).  SQLite (as well as MySQL) has non-standard typing system which makes foreign keys in Spider quite random, it also fails to enforce many of Standard SQL semantic constraints.  For example - looking at https://blog.jooq.org/a-beginners-guide-to-the-true-order-of-sql-operations/ which lists many of the sources of non-determinism in SQL execution - the list includes:

* In the presence of a GROUP BY clause, only expressions built from GROUP BY expressions (or functional dependencies thereof), or aggregate functions can be used in HAVING, SELECT, and ORDER BY clauses.
* There are cases in which GROUP BY is implied. E.g. if you write a “naked” HAVING clause
* A single aggregate function in the SELECT clause (in the absence of GROUP BY) will force aggregation into a single row
* This implicit aggregation can also be implied by putting that aggregate function in ORDER BY
* You can ORDER BY expressions that reference any columns from the FROM clause without SELECTing them. But that’s no longer true if you write SELECT DISTINCT

None of these constraints are enforced in SQLite - and all of them lead to non-deterministic output (that is, running the same query on different servers with the same schema and same content will output different outputs).

Given this observation - it would have been useful to propose a more systematic test of the Spider queries through "standard SQL validation" - including types, keys, and all forms of distinct, aggregate, order cases and correlated subqueries.

In other words, the work calls for a vigorous re-validation of the standard text-to-SQL dataset - but does not quite follow through with a systematic analysis.

[Post rebuttal]

The preliminary analysis included in the rebuttal addresses this point convincingly - I recommend the authors include a deeper version of this analysis in the final version of the paper.  This addresses this point convincingly.


B. I find the experimental analysis confusing: table 2 shows the impact of two semantic transformations (replace TOP 1 with embedded min/max queries and extend group by to dependent columns) - how many queries are impacted (up to 20%) and the execution accuracy and set match accuracy of the reference queries after these changes.  The figures are 92-97% for exec accuracy and 81-84 for exact match accuracy.  The fact that exact match accuracy is low when the query is modified should not be surprising - exact match measures query similarity.  I infer from this that only about 20% of the 20% impacted queries (about 4% altogether) are actually modified in a significant way.  Accordingly, the execution accuracy impact is 8% of the 20% impacted queries - which would be altogether 1.6% of the overall dataset.  I am not discussing here the importance of the finding - just the way it is presented which is hard to interpret.  The caption of the figure does not make it clear what the reported numbers describe and the text (425-430) is also hard to interpret.

[Post rebuttal]
The clarification of this Table in the rebuttal is satisfying.

C. Table 3 is also problematic - it shows the result of a manual analysis of 102 queries (line 453) out of the 1034 samples in the development set of Spider.  The caption of Table 3 does not mention the fact that this is sampled on only 102 queries - making the "surprising result" of the ranking reversal less convincing.

[Post rebuttal]
The clarification of this Table in the rebuttal is satisfying.

D. In the human evaluation, annotators had access to the schema and the queries - but apparently not to the results (392-398).  There is also no indication of the complexity level of the respective queries (Spider has 4 levels of complexity - simple to extra-hard).  These two factors are important to assess the reliability of the human judgment - because one can call into question the capability of programmers and non-SQL experts to assess the accuracy of a complex SQL query without looking at the execution results or experimenting with the query in an interactive manner (as supported for example in the motivation of the design of the multi-turn dataset SparC).

[Post rebuttal]
The clarification of this point in the rebuttal is satisfying.

E. The conclusion of the analysis (592-597) calls for incorporating multiple SQL queries in the dataset.  This is indeed a good direction - but another direction that should be considered is to fix the non-standard SQL queries in the dataset to make them more deterministic and more compliant.  A full analysis of leading models vs. such fixed queries over the whole dataset would be the most informative piece of data missing in this version.

[Post rebuttal]
The clarification of this point in the rebuttal is satisfying.

**Reproducibility:**

3: Could reproduce the results with some difficulty. The settings of parameters are underspecified or subjectively determined; the training/evaluation data are not widely available.

**Reviewer Confidence:**

5: Positive that my evaluation is correct. I read the paper very carefully and I am very familiar with related work.

---

> ### Author Rebuttal · Authors · 2023-08-26
>
> Thank you for your insightful review; your meticulous attention is greatly appreciated.
>
> To address the systematic analysis issue raised by the reviewer, we conducted a comprehensive analysis of the Spider, BIRD, and Spider-DK benchmarks using standard SQL validation. Specifically, we migrated the databases of these three benchmarks from Sqlite to Postgres, a renowned RDBMS known for its strict adherence to SQL standards. Subsequently, we executed all development set queries on these Postgres databases, identifying queries that produced execution errors. Here are the results:
>
> Spider dataset: Total 1034 queries
>
> UndefinedColumn: 211
> UndefinedFunction: 69
> GroupingError: 51
> SyntaxError: 4
> InvalidColumnReference: 2
>
> BIRD dataset: Total 1534 queries
>
> SyntaxError: 134
> UndefinedFunction: 103
> GroupingError: 20
> DivisionByZero: 5
> InvalidColumnReference: 4
> DatatypeMismatch: 1
> CardinalityViolation: 1
> UndefinedColumn: 1
>
> Spider-DK dataset: Total 535 queries
>
> UndefinedColumn: 80
> UndefinedFunction: 62
> GroupingError: 30
> SyntaxError: 5
> InvalidColumnReference: 2
>
> Most of the observed errors, including UndefinedColumn, SyntaxError, and UndefinedFunction, stem from differences in the accepted SQL formats between Sqlite and Postgres, necessitating modifications to align the queries with Postgres standards. For example, queries in the Spider dataset often failed due to strong typing in Postgres (e.g., comparing int with text allowed in SQLite but not in Postgres), SQLite-specific functions (e.g., strftime, iff) and Postgres not accepting literals in double quotations, interpreting them as columns.
>
> The issues related to InvalidColumnReference, GroupingError, and CardinalityViolation are primarily linked to the misuse of GROUP BY and ORDER BY statements, potentially leading to ambiguities. This experiment complements our paper's analysis, and we will be happy to add it to the paper. However, it should be noted that the benchmarks suffer from many issues not captured by a syntactic analysis, such as the problems with ties and ambiguities.
>
> Regarding the performance results in Table 2, we agree that the caption is not clear. The reported results are for the entire development set. For instance, in the Spider dataset, 206 out of 1034 queries underwent modifications. Performance evaluations considered a combined set of predicted queries, comprising 206 revised queries and 828 unchanged queries, resulting in an execution accuracy of 92.3 percent. Reviewer asked for some examples of the min/max/top transformation, below you can find some of them (we include these examples in appendix)
>
> Original Spider query:
> SELECT Name, SurfaceArea, IndepYear FROM country ORDER BY population LIMIT 1
>
> Transformed query:
> SELECT Name, SurfaceArea, IndepYear FROM country WHERE population = (SELECT min(population) FROM country)
>
> Original Spider query:
> SELECT name,  capacity FROM stadium ORDER BY average DESC LIMIT 1
>
> Transformed query:
> SELECT name, capacity FROM stadium WHERE average = (SELECT max(average) FROM stadium)
>
> Regarding Table 3, we agree that the caption should be improved. As noted, this table presents the outcome of human evaluation on a sample of 102 queries that both DIN-SQL and T5+picard methods were deemed incorrect in terms of execution accuracy. SQL experts conducted this evaluation, with 81.6% of these queries judged as correct for DIN-SQL, and only 25.5% for T5+picard. Notably, among the reference queries, only 67.3% were deemed correct. An important aspect is that annotators had access to databases to run queries and validate responses, ensuring a thorough evaluation.
>
> In the context of enhancing the text-to-SQL dataset, our proposal involves using multiple SQL queries as references for a single question rather than making the SQL queries more deterministic and compliant. This approach aligns with real-world scenarios where question and database ambiguities are common. We advocate retaining the ambiguity in questions while addressing all interpretations to facilitate a fair comparison. However, there is no reason for a non-determinism behavior of ground truth queries in the benchmarks (e.g., those attributed to GROUP BY and ORDER BY), and we strongly agree that they should be fixed.

---

### Official Review · Reviewer_ReVM · 2023-08-05

**Typos Grammar Style And Presentation Improvements:** The paper reads well.
**Soundness:** 4

**Excitement:**

3: Ambivalent: It has merits (e.g., it reports state-of-the-art results, the idea is nice), but there are key weaknesses (e.g., it describes incremental work), and it can significantly benefit from another round of revision. However, I won't object to accepting it if my co-reviewers champion it.

**Missing References:**

The citations seem adequate.

**Paper Topic And Main Contributions:**

In this paper, the authors undertake a thorough examination of the Text-to-SQL benchmarks, specifically focusing on SPIDER, SPIDER-DK, and BIRD. The study sheds light on various concerns pertaining to the evaluation of these benchmarks, particularly with regard to handling ambiguity in generating queries. By doing so, the authors aim to identify the primary reasons behind the decline in exact-set match accuracy and execution accuracies observed in state-of-the-art models.

The paper dedicates Section 5 to categorically highlight the underlying causes of inaccuracies, which include failures arising from ties in the output, ambiguity in schema matching, and erroneous assumptions about the database content. This in-depth analysis enables the readers to grasp the complexities associated with evaluating Text-to-SQL models.

To address the identified ambiguity, the authors propose insightful modifications to the development set, leveraging human annotations. The results demonstrate that state-of-the-art models perform remarkably well on these modified benchmarks, underscoring the effectiveness of the proposed changes in mitigating ambiguities.

Moreover, the paper goes beyond the evaluation aspect and sheds light on the creation of Text-to-SQL benchmarks. By outlining the issues in their inception, the authors encourage the development of more robust and comprehensive benchmarks for future research.

**Questions For The Authors:**

1. Table 2 shows results with revised SQL queries. Comparing with unrevised queries should also be shown.

2. Is query rewriting possible to automate for all types of queries or does it involve significant human expertise?



**Reasons To Accept:**

1. The paper highlights the gaps in current Text-to-SQL benchmarks and categorically highlights the issues in the benchmarks.

2. The paper would serve as an useful tool to the community in coming up with better benchmarks.

**Reasons To Reject:**

1. While the paper is useful in understanding the issues with Text-to-SQL, I was expecting modified benchmark would be shared with community. No such information is mentioned.


**Reproducibility:**

2: Would be hard pressed to reproduce the results. The contribution depends on data that are simply not available outside the author's institution or consortium; not enough details are provided.

**Reviewer Confidence:**

4: Quite sure. I tried to check the important points carefully. It's unlikely, though conceivable, that I missed something that should affect my ratings.

---

> ### Author Rebuttal · Authors · 2023-08-26
>
> We appreciate the reviewer's feedback and would like to clarify the intent and limitations of our paper. Our primary objective was to provide a comprehensive evaluation of the existing Text-to-SQL benchmarks, highlighting the inherent issues within them.
>
> Regarding the creation of a modified benchmark, we can make two comments: (1) Updating these benchmarks to address the mentioned issues necessitates a substantial human effort. (2) Many benchmarks in the Text-to-SQL domain, such as Spider and BIRD, incorporate holdout test sets for official leaderboards and comparisons of text-to-SQL methods. Our access is limited to the development and training sets of these benchmarks, and we are unable to modify test sets. Thus, even updating the development and training sets alone would not fully resolve the underlying problems within these benchmarks, since the final performance comparison would be reported on the test sets with inherent issues.
>
> In essence, our paper's aim was to draw attention to the issues within existing benchmarks rather than proposing a new benchmark. We believe that the responsibility for benchmark updates rests with the benchmark creators, and we encourage them to consider our findings when revising their benchmarks to ensure a more accurate evaluation of Text-to-SQL methods in the future.
>
> In Table 2, we presented the execution accuracy and exact set match accuracy of the revised queries, using the unrevised queries as references. The purpose of this comparison was to highlight that even with correct SQL queries manually crafted by experts, we cannot achieve 100% accuracy on the benchmark tasks. In such a scenario, comparing the results with unrevised queries would not provide meaningful insights, as both exact set match accuracy and execution accuracy would be 100% for unrevised queries.
>
> Automating query rewriting faces inherent challenges, particularly when dealing with failures stemming from schema ambiguity, erroneous assumptions about the database content, and the ambiguity of natural language utterances. These challenges arise because there is no specific structure to address the failures systematically. Successful query rewriting in these cases necessitates a deep understanding of table and column semantics to identify ambiguities and erroneous assumptions. In cases of ambiguity, human expertise is essential to disambiguate the context, as these situations often lack clear guidelines. Detecting erroneous assumptions often involves introducing new data to the database and meticulously reviewing and correcting failed queries on a case-by-case basis.
>
> We hope this clarifies our paper's scope, and we welcome any additional suggestions or questions from the reviewer.

---

### Meta-Review · Area_Chair_ShQD · 2023-09-18

**Recommendation:** 4

**Metareview:**

The reviewers liked how the paper highlights current issues with existing text-to-sql benchmarks through experiments and human analysis. The paper might have created expectations for new benchmarks and evaluation metrics that it did not deliver, but also never claimed to. Other concerns have been addressed in an exemplary fashion by the authors during the rebuttal.

---

### Decision · Program_Chairs · 2023-10-07

**Decision:**

Accept-Main

**Comment:**

The reviewers liked how the paper highlights current issues with existing text-to-sql benchmarks through experiments and human analysis. The paper might have created expectations for new benchmarks and evaluation metrics that it did not deliver, but also never claimed to. Other concerns have been addressed in an exemplary fashion by the authors during the rebuttal.